# Stability follows efficiency based on the analysis of a large perovskite solar cells ageing dataset

Noor Titan Putri Hartono [1,3] ✉, Hans Köbler [1,3], Paolo Graniero [1,2], Mark Khenkin [1], Rutger Schlatmann [1], Carolin Ulbrich [1] & Antonio Abate[1] ✉

While perovskite solar cells have reached competitive efficiency values during the last decade, stability issues remain a critical challenge to be addressed for pushing this technology towards commercialisation. In this study, we analyse a large homogeneous dataset of Maximum Power Point Tracking (MPPT) operational ageing data that we collected with a custom-built High-throughput Ageing System in the past 3 years. In total, 2,245 MPPT ageing curves are analysed which were obtained under controlled conditions (continuous illumination, controlled temperature and atmosphere) from devices comprising various lead-halide perovskite absorbers, charge selective layers, contact layers, and architectures. In a high-level statistical analysis, we find a correlation between the maximum reached power conversion efficiency (PCE) and the relative PCE loss observed after 150-hours of ageing, with more efficient cells statistically also showing higher stability. Additionally, using the unsupervised machine learning method self-organising map, we cluster this dataset based on the degradation curve shapes. We find a correlation between the frequency of particular shapes of degradation curves and the maximum reached PCE.

Perovskite solar cells (PSCs) have reached a competitive efficiency of 26.1%[1], indicating that the technology has the potential to be commercialised and implemented on a large scale. However, the current PSCs lifetime is subpar (~1 order of magnitude lower) compared to silicon solar cells[2], even if environmental stressors like water and oxygen are excluded.

Several publications have intended to gain insights into PSC stability via machine learning[2–4]. The recently published Perovskite Database Project[5], which collected > 42,400 PSCs data extracted from literature, is a major step towards machine learning application in the PSCs field. Despite the database's large size, less than 20% of data points (~7500) have any degradation data available, which is also not necessarily homogeneous. Graniero et al. have shown that the degradation data in this database has low quality to apply supervised machine learning algorithms properly[6]. The study points out that rather than adding more low-quality data points, a higher data quality (i.e. with more complete information) is needed to train machine learning algorithms. Despite this issue, Zhang et al. recently performed a statistical analysis on the Perovskite Database Project by introducing a new figure of merit for stability called $T_{S80m}$. They project the measured $T_{S80}$ (the time taken to reach 80% of the stabilised efficiency at the end of the burn-in region)[3] to $T_{S80m}$, which is the predicted value under reference conditions (300 K, 20% relative humidity, and 1 sun illumination). The authors re-calculate $T_{S80}$ with the help of acceleration factors which are determined based on various assumptions and which consider the temperature, humidity, and illumination levels during the actual ageing test[7]. While this choice enables a more rigorous statistical analysis, the uncertainty from co-dependencies between different stressors and the range of parameters lowers the accuracy of

[1]Helmholtz-Zentrum Berlin für Materialien und Energie, 14109 Berlin, Germany. [2]Department of Business Informatics, Freie Universität Berlin, 14195 Berlin, Germany. [3]These authors contributed equally: Noor Titan Putri Hartono, Hans Köbler. ✉e-mail: titan.hartono@helmholtz-berlin.de; antonio.abate@helmholtz-berlin.de

calculated $T_{S80m}$ and the analysis does not consider the shapes of degradation curves.

Here, we present a statistical analysis on a highly homogeneous dataset of maximum power point tracking (MPPT) ageing curves, collected during the past 3 years under controlled environmental conditions in a custom-built High-throughput Ageing System[8]. We analyse the power conversion efficiency (PCE) loss after 150 h relative to the maximum efficiency reached during the ageing test. We consider the maximum efficiency to reflect the maximum potential or capability of the solar cell under test, which is why it is chosen as a figure of merit. The dataset comprises MPP-tracks of devices of various structures, including lead-halide perovskite absorbers (both organic and inorganic), charge-selective layers (small molecules, polymer, and inorganic), diverse contact layers (silver, copper, and gold), and architectures (both n-i-p and p-i-n). See Supplementary Note 1: Data Quality, Supplementary Tables 1 and 2 for a detailed description.

In addition, since PSCs show large variations in degradation curve shapes, there is still no universal metric for PSC stability. Our dataset shows various degradation curve shapes that can be categorised into different groups (see Supplementary Fig. 1). In this study, we perform an unsupervised machine learning method called self-organising map (SOM) to obtain degradation curve shape clusters from the MPP-tracking dataset, where we are able to identify the dominant shape of ageing curves. We observe a correlation between the occurrence of particular shape clusters and the maximum reached PCE. We believe that categorising ageing test data and identifying the main shapes will bring PSC research closer to finding suitable lifetime metrics for PSCs.

## Results and discussion
### Dataset description
We collected 2245 MPPT curves of various device architectures, layers, and perovskite composition materials from August 2019 to August 2022 in the HySPRINT Laboratory at Helmholtz-Zentrum Berlin, Germany. The ageing tests were performed in a custom-built ageing setup[8] under continuous illumination at 1 sun and with individual MPP-tracking for every single solar cell. Variations of the ageing conditions are the device temperature and the use of a UV filter. A summary of the ageing conditions is shown in Supplementary Information Table 1. All devices were fabricated within the same laboratory, albeit fabricated by different researchers (total: ~33 researchers). Within the dataset, 502 solar cells have an n-i-p architecture and 1,743 solar cells have a p-i-n architecture. The devices have different electron transport materials (e.g. $TiO_2$-c, $C_{60}$/BCP, PCBM, and others), hole transport materials (e.g. spiro-OMeTAD, MeO-2PACz, NiO, PTAA, and others), and top electrodes. The perovskite absorber material dominating the dataset is the so-called triple cation (3CAT) perovskite[9] with the general formula $Cs_xMA_yFA_zPbI_mBr_n$ (881 cells, with $x+y+z=1$ and $m+n=3$); other absorbers present in the dataset include $CsPbI_3$ (218 cells) and $FAPbI_3$ (56 cells). A complete breakdown of the cell numbers based on the material layers is shown in Supplementary Information Table 2.

### Cell grouping and relative change in PCE
Only the first 150 h of all ageing experiments are analysed to have the maximum amount of comparable data (see Supplementary Fig. 2 for degradation data length distribution in this dataset), and data points which reach maximum efficiency beyond 150 h are excluded (see Supplementary Note: Degradation Time Length for a detailed description). To investigate trends in dependency of the maximum power conversion efficiency (PCE), the ageing tracks are grouped based on the maximum PCE reached during the first 150 h of MPP-tracking as follows: <10.4%, 10.4–14.2%, 14.2–16.8%, 16.8–19.2%, and >19.2%. The data points are divided so that every maximum PCE group has an equal number of cells (449 cells/ group). Maximum PCE is chosen as a parameter of interest because many devices undergo an initial phase of grave losses in PCE at the beginning of the experiment,

also known as the burn-in period[3]. In some cases however, devices undergo an initial gain in efficiency, so-called light soaking improvements[3] and as a result, their initial PCE will not compare well to devices for which the MPP-track immediately decays and for which the initial PCE equals the maximum PCE. Consequently, the maximum reached PCE is selected as value of interest since it reflects the maximum potential or capability of the solar cell under test more universally between different ageing curve shapes. The change of PCE observed after ageing for 150 h relative to the maximum PCE in a cell ($\Delta PCE$, rel) is calculated for each of the different efficiency groups, according to Eq. 1.

$$\text{Relative change in PCE} (\Delta PCE, rel) = \frac{\text{Maximum PCE} - \text{PCE} (150\,h)}{\text{Maximum PCE}}$$

(1)

This figure of merit reflects the percentage of PCE loss after 150 h concerning the maximum efficiency reached. Note that the relative change $\Delta PCE$, rel will include the 'burn-in' phase if the MPP-track shows an immediate decay (and the maximum PCE equals the initial PCE), while it excludes the meta-stable phase if the MPP-track shows an initial gain where the maximum is reached after the stabilisation phase. However, $\Delta PCE$, rel should be understood as the relative loss observed after 150 h of ageing time with respect to the maximal capability of the system. See Supplementary Note: MPPT Behaviour and Supplementary Fig. 3 for a detailed discussion. The relative change $\Delta PCE$, rel is plotted over the efficiency groups in Fig. 1b.

An observation in Fig. 1b shows that as the maximum PCE reached increases, the mean of the $\Delta PCE$, rel decreases. This trend suggests that solar cells reaching a higher maximum efficiency during the ageing experiment, statistically also offer a lower loss in efficiency in the first 150 h and can be considered to be more stable according to this figure of merit for stability. The trend is supported by the linear regression performed on the mean of the maximum PCE group and the mean of the $\Delta PCE$, rel shown in Fig. 2. Based on the linear regression, for every 1% increase in maximum PCE reached during the first 150 h of degradation, the $\Delta PCE$, rel is reduced by -1.5%.

We point out that the observation that more efficient devices also provide higher stability should not be taken literally as a design rule. Specific device layers are known to improve PCE at the expense of long-term stability and vice versa (e.g. inorganic perovskites, spiro-OMeTAD as hole transport layer, and carbon electrodes). The statistical statement given here only illustrates the general trend when looking at the wide range of devices with various stacks and, therefore, different degradation mechanisms. Yet, when looking across different PSCs with various intertwined degradation mechanisms, we see a clear trend that more efficient cells are statistically more likely to have a longer lifetime. Understanding the underlying reasons behind such a trend can strengthen our understanding of PSC ageing behaviour and accelerate the development of highly stable devices.

We propose two hypothetical explanations for the observed relationship between PCE and stability with different causality.

Firstly, at the high level, statistically better stability of high-efficiency PSCs might be explained using a simple conservation of energy model, which states that the total energy of a system stays constant over time. A perovskite solar cell could be modelled as a system that converts incident solar into electric power, as illustrated in Fig. 3. The lower the efficiency of a solar cell, the larger is the amount of energy that remains in the system.

Generally, unavoidable processes lowering the power output of a single junction solar cell are Carnot, Boltzmann and emission losses, thermalisation of above-band-gap photons, and transmission of below-band gap photons[10]. According to the detailed balance limit of efficiency by Shockley–Queisser[11], this leaves ~43–48 % of the input energy

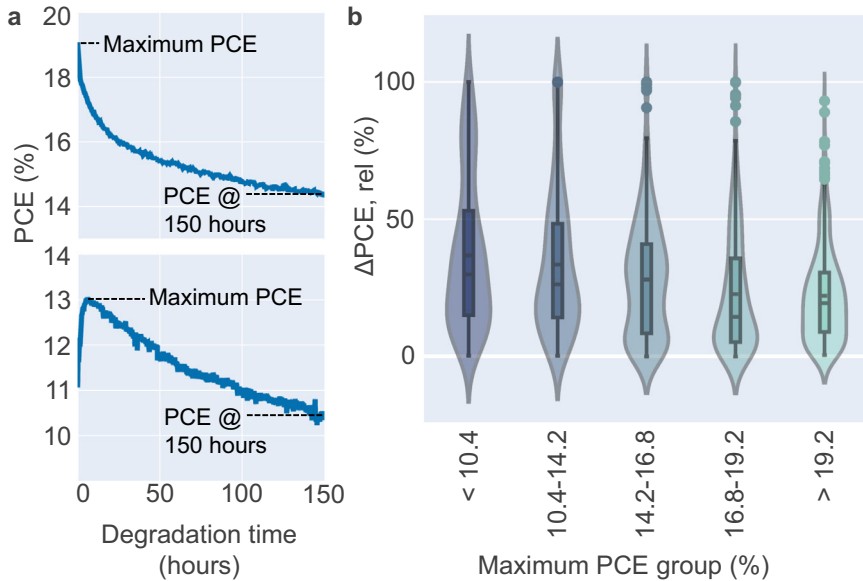

**Fig. 1 | Statistical analysis of a large MPPT-ageing dataset. a** Schematic on how the relative change between maximum PCE and PCE at 150 h (*ΔPCE, rel*) is calculated. **b** The *ΔPCE, rel* for the five groups of maximum PCE. The solid line in the box represents the median, the dashed line in the box represents the mean, the error bar represents the corresponding interquartile range (between the 25th and 75th percent quartile), and the shaded area represents the distribution of the data points for the group.

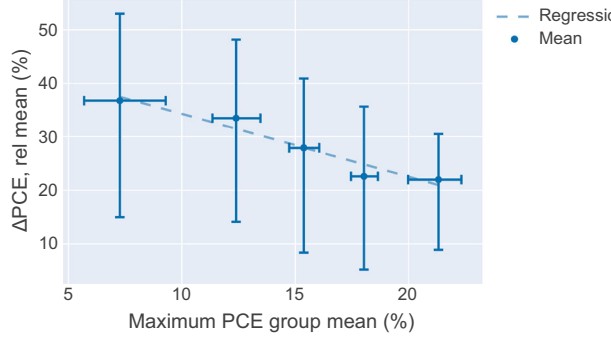

**Fig. 2 | Regression fit of the maximum PCE group mean.** The relationship between the mean for each maximum PCE group and its respective mean of the *ΔPCE, rel* is shown. The dashed line is the regression fit of the mean points. The error bar for both axes represents the interquartile range between the 25th and 75th percentile of the maximum PCE group and the *ΔPCE, rel* data.

to be absorbed, and a maximum ~30% of the input energy to be converted into electric energy by a solar cell under AM1.5 G for a band gap of 1.6–1.7 eV[9] corresponding to the triple cation perovskite which provides the majority of data points of the present data. Assuming an equal or similar band gap and similar optical properties, the differences between a low and a high efficient solar cell originate in recombination[12] or transport losses[13]. The fraction of absorbed energy, which is not extracted again from the device electrically, remains in the solar cell and gets dissipated in the device where it can potentially trigger degradation. For example, in a 10% efficient "triple cation perovskite (energy absorbed ~466 W/m²) device, ~366 W/m² of the input energy would be potentially available to cause damage under 1 sun, while for a 20% device it would be ~266 W/m². For an estimation of the influence of the band gap in this dataset[9, 14–21] on this theory, see Supplementary Information: Influence of band gaps, Supplementary Table 3, Supplementary Figs. 4 and 5. In the case of trap-assisted recombination, which is likely to be non-radiative, the energy would be transferred to heat that can initiate or accelerate degradation

mechanisms in the device. If the device efficiency is lowered by transport limitations, excess charges will remain in the device and can potentially trigger degradation processes. For example, Lin et al. have shown that the insufficient extraction of generated charge carriers leads to a stimulation of ion migration causing enhanced degradation of devices[22] and Di Girolamo et al. reported that injected charges induce phase separation[23].

The theory that excess energy might be responsible for instability is also in accordance with the observation that $V_{MPP}$ is the mildest electronic load ageing condition for PSCs[24] and also other PV technologies[25]. The reasoning is that at MPP, the maximum possible amount of electric energy is extracted from the device, leaving less energy or charges in the device potentially triggering degradation, which leads to the higher stability observed under MPP-tracking compared to $J_{SC}$ (short-circuit current) and $V_{OC}$ (open-circuit voltage) conditions. At $J_{SC}$ and $V_{OC}$ conditions, no electric energy is extracted from the system and all absorbed energy is potentially available to trigger degradation processes. Note that under $V_{OC}$ condition additional effects might come to play[25–27].

We want to point out that our high-level statistical analysis is performed over various PSCs with a wide range of material layers used. Hence it is very likely that also various underlying physical degradation mechanisms are present and it is impossible to directly relate the results of the analysis to the physical causes of degradation. Therefore, the given theory of the conservation of energy should also be understood as a high-level explanation.

A second potential explanation for the correlation between efficiency and stability may lie in the presence of pinholes and defects, incomplete solvent removal, or generally poor device quality. This would put some devices at a disadvantage, and the "fresh device" is already defective at the start of the ageing test, lowering the PCE. Degradation mechanisms could be triggered at defect sites or unremoved solvent. Therefore, the presence of imperfections of any origin might affect efficiency and stability simultaneously. In this potential explanation, the causality is that the same reasons that make a device low in efficiency is also causing a device to be unstable, while in the energy conversion model the low efficiency would make devices unstable as a secondary cause.

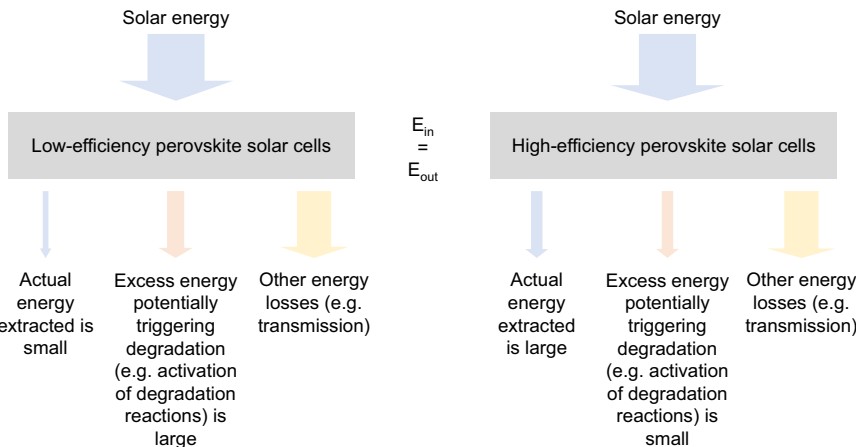

**Fig. 3 | The energy conservation schematic for the perovskite solar cell system.** The incident solar power coming into the system equals the power coming out. In the case of high-efficiency cells, the actual power extracted and converted into electricity is relatively large, leaving a smaller amount of energy to potentially trigger degradation.

## Degradation curve shape clustering

We observe large variations of degradation curve shapes in the first 150 h of the degradation process (see Supplementary Fig. 1). Hence, a machine learning method called self-organising map (SOM) was implemented to find the main degradation curve shapes in the dataset. SOM is a popular unsupervised machine learning method for clustering data by converting the nonlinear statistical relationships of high-dimensional data into geometric relationships in low-dimensional nodes while preserving the topological structure of data, i.e. the relations between one data point and others[28]. In short, SOM compresses information of high-dimensional data, providing visualisation and abstraction of the data.

SOM is utilised to identify the distinct types of degradation curves' shapes based on the normalised PCE data. Four clusters of curve shapes are identified and shown in Fig. 4a: initial gain (cluster 1, total: 1324 points), slow exponential decay (cluster 2, total: 722 points), medium exponential decay (cluster 3, total: 237 points), and fast-exponential decay (cluster 4, total: 97 points). Four is the optimum number of clusters based on the elbow plot analysis shown in Supplementary Fig. 6 and discussed in Supplementary Information: SOM Quantisation Error. In this case, the SOM parameters used explicitly are sigma = 0.5 and learning rate = 0.1.

While it is known that perovskite solar cells can show very diverse ageing behaviours[3] (see also Supplementary Fig. 1 for a higher number of clusters), this is an impressive example of how diverse ageing tracks are in reality. The vast deviations between the cluster shapes clearly show that the commonly used device lifetime metric $T_{80}$ (among others) cannot be considered universal for perovskite solar cells. A new figure of merit or a set of parameters that can work across different shapes of degradation curves is therefore urgently needed to compare the ageing results of PSCs.

The normalised count of different clusters for each maximum PCE group is shown in Fig. 4b. As the maximum PCE reached increases, cluster 1, initial gain, increases in share and becomes the majority (> 50% of the share) of degradation curves' shapes for higher maximum PCE groups, showing that more optimum devices tend to have such ageing curves. Simultaneously, the fraction of degradation curves with cluster 4, fast-exponential decay decreases, meaning that high-efficiency devices show less of this type of failure. Those trends can still be observed when the clustering is performed with other possible SOM parameter values (see Supplementary Fig. 7). The clustering results generated by SOM also agree with the result generated with the k-means clustering method[29] (see Supplementary Fig. 8). Moreover, the slight dominance of cluster types in

dependency of the architecture (majority of p-i-n architecture devices have initial gain shape, and majority of n-i-p devices have exponential decay shapes) are in agreement with an earlier single-stack report by Saliba et al.[30] (see Supplementary Information: Device Architecture Impact on Clustering).

In Fig. 5, the relative change in efficiency $\Delta PCE, rel$ is shown in dependency of the maximum PCE group and sorted by cluster type (a table with the statistics is provided in Supplementary Table 4). The histogram in accord with Fig. 5 is also available in Supplementary Fig. 9. Cluster 4 fast-exponential decay clearly has the largest $\Delta PCE, rel$ with most values located around 100% relative change (i.e. fully degraded). Within the same cluster 4 fast-exponential decay, no data point comes from > 19.2% maximum PCE group, highlighting again that higher efficient devices do not show this catastrophic failure. Defective fresh devices could be one of the reasons that some of the cells, especially from cluster 4, exhibit an instant failure behaviour within the first few hours of the ageing test.

We can also observe that the distribution of $\Delta PCE, rel$ in dependency of maximum PCE shown in Fig. 1b possesses contributions from different clusters and is composed of the combination of those. This is most apparent in the maximum PCE group of <10%, where all types of clusters are represented and a large spread in the distribution is seen in Fig. 1b. A possible outcome of this analysis is that in future ageing tests, the curve shape might be a predictor of stability: If we can observe the initial gain curve shape during the first couple hours, devices have a higher likelihood to be stable within the first 150 h.

In this study, we analysed a large dataset of MPPT-ageing data collected in-house. The specific aim was to investigate the relationship between the maximum PCE reached during the first 150 h of operational MPPT ageing and the relative loss after 150 h $\Delta PCE, rel$ as a metric for PSCs stability. We discovered that the higher the maximum PCE reached in the first 150 h of testing is, the lower is the mean of $\Delta PCE, rel$. While this statistical relationship cannot be generalised, it is an encouraging finding that efforts in improving PSC's efficiency go alongside enhancing stability. The finding could be explained using an energy conservation model. Another possible explanation is that increased defect densities render devices to have low efficiency and simultaneously lower stability, as defects might also act as initialisation points for degradation.

Secondly, we clustered the MPPT curves regarding their shape and we discovered that the degradation curve shape is also related to both the cells' maximum PCE group and stability. The initial gain degradation curve shape delivers lower $\Delta PCE, rel$ than other shapes, and this shape type is seen more frequently on cells with higher

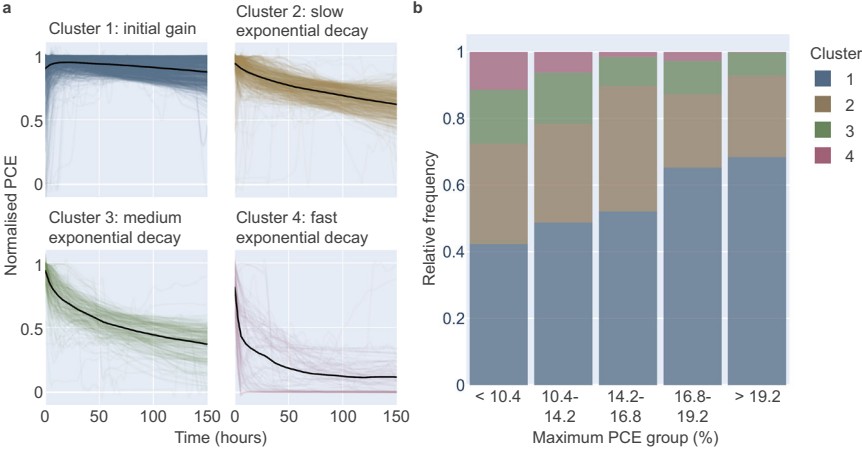

**Fig. 4 | Degradation curve shape clustering. a** The four different clusters of PSCs' degradation curve shapes categorised through the self-organising map (SOM) method (parameters: sigma = 0.5, learning rate = 0.1). Cluster 1: initial gain, cluster 2: slow exponential decay, cluster 3: medium exponential decay, and cluster 4: fast-exponential decay. The black lines represent the centre average of the curves belonging to each cluster. **b** The relative frequency of each cluster for a given maximum PCE group represented in a bar plot. The violet box represents the fast-exponential decay shape (cluster 4), which decreases significantly in occurrence for higher maximum PCE groups, while initial gain shape (cluster 1) can be observed more frequently for more efficient devices.

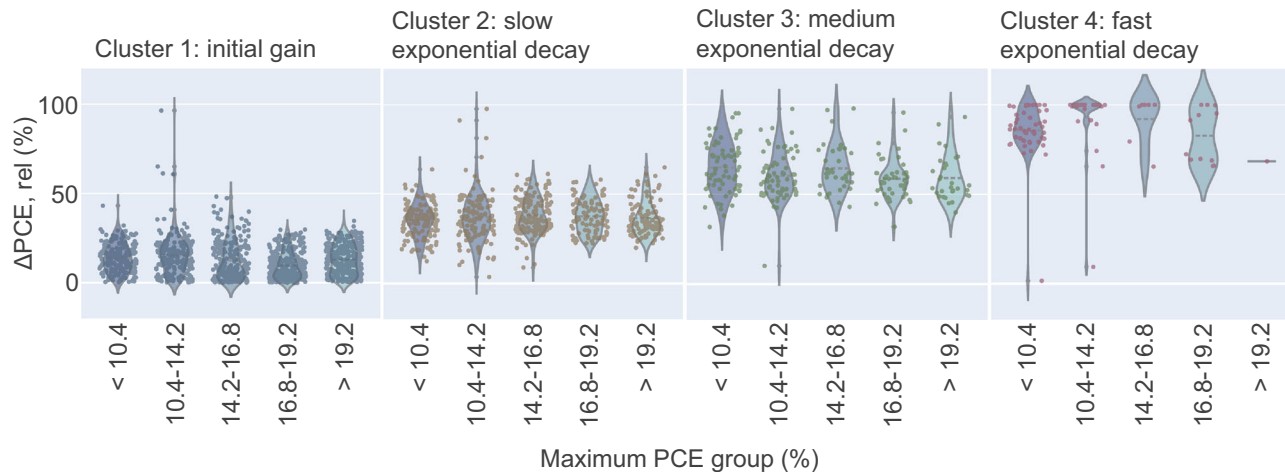

**Fig. 5 | The distribution of *ΔPCE, rel* based on clusters.** The data points are grouped into previously calculated clusters, and their respective maximum PCE group. In the fast-exponential decay group (cluster 4), there is no data point for the >19.2% maximum PCE group.

maximum PCE. This might act as an early indicator for stability: if the initial gain curve shape is observed during the first couple of hours, devices have a higher likelihood to be stable during the first 150 h.

## Methods

### Ageing of solar cells

Solar cells were aged in a custom-built High-throughput Ageing System[8]. Special electronics were used to MPP-track all cells individually. A perturb and observe algorithm[31] with a delay time of 1 s and a voltage step-width of 0.01 V was applied to track the MPP. $PCE_{MPP}$ values were taken every 2 min for all cells automatically. Devices were constantly kept at 25, 45, 65 or 85 °C (see Supplementary Table 1) by the use of actively controlled Peltier-elements. Active areas of devices were touching a heat pad for direct thermal coupling. Ageing was performed under a continuous flow of nitrogen (in some cases air) in a closed box, no additional encapsulation was used in the majority of samples (2220). Twenty-five cells were encapsulated with a glass-to-glass encapsulation. Sunlight with 1 sun intensity was provided by a metal-halide lamp using a H6 filter. Supplementary Fig. 10 shows the spectrum of the light source in comparison to AM1.5 G. The light intensity was actively controlled with the help of a silicon irradiance-sensor which was calibrated using a KG3-filtered silicon reference cell certified by Fraunhofer ISE. In 475 cells, a UV filter was used to block UV-light with wavelengths below 380 nm. The ageing test conditions are summarised in Supplementary Table 2 in Supplementary Information. Tests are in accordance with the protocols ISOS-L-1I or ISOS-L-2I[3].

### Data analysis

All the data pre-processing and analysis steps are done in Python. MPPT data pre-processing involves resampling to 10 minutes (since the frequency of measurement varies for each data point), interpolation using the Akima method[32], normalisation based on the maximum point of the data (MaxAbsScaler) using scikit-learn[33] package, and applying a Savitzky-Golay filter[34] to reduce the noise within the dataset using SciPy[35] package (with window length parameter = 71).

The MaxAbsScaler normalisation is performed on the data by dividing the MPPT PCE across time with the maximum MPPT PCE reached in the first 150 h of ageing test (Eq. 2).

$$x_{\text{MaxAbsScaler}} = \frac{x}{\max(|x|)} \tag{2}$$

The self-organising map[28] analysis is done using the MiniSOM[36] package, and the linear regression and *k*-means clustering is performed using the scikit-learn[33] package. The parameters for each machine learning model are stated under the respective results in the manuscript.

## Data availability
The MPPT data used in this study are available in the Zenodo database under accession code 8185883 [37].

## Code availability
The Jupyter Notebook with the code for running the analysis is available on Zenodo under accession code 8181602 [38] and Github.

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

## Acknowledgements

H.K. acknowledges the VIPERLAB project funded by the European Union's Horizon 2020 research and innovation programme under grant agreement N°101006715. P.G. acknowledges the support of the Helmholtz Einstein International Berlin Research School in Data Science (HEIBRiDS). M.K. acknowledges the support of European partnering protect TAPAS (PIE-0015). C.U. acknowledges support by the Helmholtz Association under the programme "Energy System Design". All authors acknowledge the researchers who have shared their PSCs data: Ghefar Adwan, Ece Aktas, Amran Al-Ashouri, Laura Canil, Pietro Caprioglio, Janardan Dagar, Carola Ferber, Emilio Gutierrez-Partida, Juanita Hidalgo, Zafar Iqbal, Jose Jerónimo-Rendón, Ivona Kafedjiska, Eike Köhnen, Bor Li, Guixiang Li, Jinzhao Li, Xin Li, Silvia Mariotti, Natalia Maticiuc, Jorge Pascual, Nga Phung, Manuel Piot, Marcel Roß, Rajarshi Roy, Florian

Scheler, Silver-Hamill Turren-Cruz, Qiong Wang, Fengjiu Yang, Jorge Pascual, Thomas Gries.

## Author contributions

H.K. collected the degradation measurement data, including the cells' parameters and fabrication data and provided devices for ageing. N.T.P.H, H.K., and P.G., performed data cleaning. N.T.P.H. performed data pre-processing and machine learning analysis. M.K., R.S., C.U., and A.A. conceptualised and supervised the research. N.T.P.H, H.K., and A.A. wrote the first version of the manuscript. All authors participated in work planning, reviewing, and editing the manuscript.

## Funding

## Competing interests

The authors declare no competing interests.
