## [Peer Review File · Nature Communications]

REVIEWER COMMENTS

Reviewer #1 (Remarks to the Author):

In this study, the authors collected a large MPPT-ageing dataset with a custom-built ageing system, clustered the MPPT curves with the SOM machine learning method, and investigated the relationship between maximum PCEs, MPPT-curve shapes, and relative PCE loss. Interestingly, the degradation curve shape and the relative PCE change are related to the maximum PCE for the results, and the curve shape might act as an indicator of stability. However, there are still some concerns and missing parts that may affect the final results. I recommend it for publication after major revisions. Please find below the specific comments and suggestions.

1. In Equation 1, the authors chose the maximum PCE rather than the initial PCE as the parameter to exclude the effect of the burn-in period, and the indicator ΔPCE_{rel} is similar to the slope of the linear ageing part as the author said. However, for devices that begin with a rapid decay, the maximum PCE equals to the initial PCE, the burn-in period is involved, and ΔPCE_{rel} is obviously larger than the slope. This may lead to the conclusion that devices with initial gain are more stable. The authors should give a detailed discussion, or modify the indicator if possible.

2. In Equation 1, the duration of the burn-in period will affect the value of ΔPCE_{rel} . For example, if the initial gain period is the majority or longer than 150 hours, ΔPCE_{rel} cannot represent the device stability accurately. How much does it affect the conclusion? The authors should give a detailed discussion about it, or modify the indicator if possible.

3. In Figure S3, what does the quantization error mean and how is it calculated? How is the optimum value $n=4$ determined?

4. In Figure 4, the degradation curve shapes are categorized into four different clusters containing one initial gain cluster and three exponential decay clusters. In some previous reports (10.1016/j.joule.2018.05.005 & 10.1038/s41560-019-0529-5 & 10.1038/s41560-017-0060-5), the burn-in period is independent of the linear degradation period and reversible, and the shape may be affected by device structures (initial decay for nip and gain for pin). The authors should check whether degradation curve shapes are related to device structures.

5. The authors could also add a discussion on whether different degradation rates of the three exponential decay clusters are related to the testing conditions (temperature and atmosphere) since the device stability is sensitive to them.

Reviewer #2 (Remarks to the Author):

Machine learning, one of artificial intelligence methods is useful in many fields. In this manuscript, the relationship between efficiency and stability of perovskite solar cells (PSCs) is explained from a novel perspective, i.e. using the unsupervised machine learning method self-organising map. Unsupervised clustering is achieved according to data topology information. Moreover, the clustering results are good, which can support the conclusions. The novelty of this work is high compared with the traditional additive strategy in PSCs. Therefore, I suggest publication. However, this paper is preliminary and needs major revision. The suggestions are followed:

1. First of all, this paper is not well organized. Too many details should be put into supporting information such as Dataset description. The authors should give logical description in the Dataset description: how we get higher data quality? Tell the readers how high the data quality used in this paper. By the way, is artificial intelligence used for the paper writing, such as ChatGPT?
2. This manuscript only compared the results of SOM and k-means, as well as analysis of different k

values of k-means. However, it lacks direct analysis on the number of SOM clusters.

3. The figures are not satisfactory since researches should consider the art of science. Figure 1(a) should be from the actual experimental data. What is the exact value in the longitudinal coordinate in Figure 1a?

4. The theory from Figure 3 is common sense. Authors should consider the detailed physical process in perovskite solar cells. How excess energy triggered the degradation in physics? This will help readers to connect machine learning results with the physical process in perovskite solar cells.

Reviewer #3 (Remarks to the Author):

Synopsis

"Stability Follows Efficiency: Analysis of a Large Perovskite Solar Cells Ageing Dataset" by Hartono et al. performs interesting statistical analyses on degradation data collected from several thousand perovskite solar cells over a period of years. The work highlights several interesting correlations between the initial efficiency performance of solar cells, the rate at which they degrade, and the "shape" with which their PCE vs. time curves change. Rather than relying on any specific information about the cells, like their architecture, transport layers, absorber, testing temperature, or even sub-degradation type (FF, J_{sc} , Voc) to understand how and why they degrade, the work focuses purely on high-level abstraction of trends across thousands of cells that vary in essentially every way that perovskite solar cells can vary. Intuitively, I suspect that by zooming in on the degradation difference between any pair of solar cells in the work, the reasons why they degrade differently would be apparent from things like material choice. However, I think there is value also in this high-level approach, since it may help the community identify larger trends. Thus, after the points below are addressed, I think the work is suitable for publication in Nature Communications.

Major Points

1. The definition of TS80m is not clear from the description provided from lines 53-55. The definition of Ts80 is clear, but in saying "...normalizes an extrapolated TS80 ... to a reference condition...", it is unclear what normalization is being performed. Normalizing to efficiency at a reference condition? Using an acceleration-factor like analysis to predict lifetime at the reference condition? This should be clarified.

2. On lines 88-89, the average number of batches contributed per researcher (or a distribution of batch numbers) is also of interest. Since the "minimum" number of solar cells per batch is 48, and the average number of solar cells tested per researcher is 72, this description could be interpreted as many (or most) researchers contributing just one large batch to test stability, or as many (or all) researchers contributing just a few cells per batch for lifetime testing in which case each researcher would have cells from perhaps 10+ batches on average being tested.

3. Since care is given to define Maximum PCE (which may either arrive at $t = 0$, or slightly after in the case of cells with a burn-in), similar care should be given to Minimum PCE over the course of the test. Equation 1 is defined for a non-general case, where the Minimum PCE universally occurs at the end of the testing period explored (150 hr). However, if a cell increases in PCE and does not decay within the 150 hr window, then Minimum PCE would occur at $t = 0$ rather than $t = 150$ and thus the relative change in PCE could be written more generally as $(\text{Maximum PCE} - \text{Minimum PCE}) / \text{Maximum PCE}$. Perhaps all of the solar cells within the data set decay continuously and thus reach their lowest point at 150 hr, but if this is the case, it should be stated explicitly.

4. The concept described in lines 160 – 168 that more efficient cells dissipate energy in positive ways, and thus leave less excess loss energy to drive degradation is an interesting concept. However, the given dataset would need considerably more information to justify this speculation. In particular, it is

more interesting to know what percentage of the absorbed energy is converted into electricity rather than the percentage of incident energy (which is PCE, by definition). Otherwise, differences in PCE could be a function solely of absorbing more energy, even if the conversion efficiency of absorbed power is the same across device architectures and efficiency groups. For example, if the internal conversion efficiency of all cells was 70%, and differences in PCE were solely from absorption, the higher PCE cells would actually have a larger amount of excess energy remaining in the cell than the lower efficiency cells. An estimation or measurement of the internal conversion of the tested cells would be an interesting addition to the work.

5. The statement beginning on line 175 that at JSC and VOC “no energy is extracted from the system” should be more nuanced, since by thermodynamic definition, ~100% of the energy is extracted from the system at JSC and VOC, but is done via re-radiation and thermal conduction rather than by providing an external circuit with electrical power. The discussion from lines 170 to 184 should also include the effect that voltage itself may have on perovskites.

Minor format and style suggestions

1. The abstract at least in the version I saw is not a continuous paragraph. There are two newline commands. It should be a continuous single paragraph

2. On line 45, “only” and “less than” are redundant, use one or the other.

3. The use of et al is applied inconsistently—for example, Graniero et al uses only the author’s family name, while Zhuang Zhang et al. uses both given and family name.

4. “High-throughput Ageing System” need not be capitalized on line 62, unless it is a specific product or system with a title.

5. On line 97, the formula for the 3CAT perovskite should include the stoichiometry co-dependencies of x, y, z, m, and n (e.g. $CsxMAyFAzPbImBrn$, where $x + y + z = 1$ and $m + n = 3$)

REVIEWER COMMENTS

Reviewer #1 (Remarks to the Author):

In this study, the authors collected a large MPPT-ageing dataset with a custom-built ageing system, clustered the MPPT curves with the SOM machine learning method, and investigated the relationship between maximum PCEs, MPPT-curve shapes, and relative PCE loss. Interestingly, the degradation curve shape and the relative PCE change are related to the maximum PCE for the results, and the curve shape might act as an indicator of stability. However, there are still some concerns and missing parts that may affect the final results. I recommend it for publication after major revisions. Please find below the specific comments and suggestions.

1. In Equation 1, the authors chose the maximum PCE rather than the initial PCE as the parameter to exclude the effect of the burn-in period, and the indicator ΔPCE_{rel} is similar to the slope of the linear ageing part as the author said. However, for devices that begin with a rapid decay, the maximum PCE equals to the initial PCE, the burn-in period is involved, and ΔPCE_{rel} is obviously larger than the slope. This may lead to the conclusion that devices with initial gain are more stable. The authors should give a detailed discussion, or modify the indicator if possible.

We thank the reviewer for this truly valuable comment. It is true that in the case of decay, the maximum PCE equals the initial PCE and the stabilisation phase is accounted for in ΔPCE_{rel} . In the case of initial gain, maximum PCE does not equal the initial PCE and ΔPCE_{rel} mainly considers the linear part of the curve. We have removed the statement that ΔPCE_{rel} is similar to the slope of the linear ageing part and added a detailed discussion of the parameter in **Supplementary Information: MPPT Behaviour** (page 6) to clarify this.

2. In Equation 1, the duration of the burn-in period will affect the value of ΔPCE_{rel} . For example, if the initial gain period is the majority or longer than 150 hours, ΔPCE_{rel} cannot represent the device stability accurately. How much does it affect the conclusion? The authors should give a detailed discussion about it, or modify the indicator if possible.

The Reviewer is asking an important issue regarding the initial gain period of the ageing curves. We added a detailed discussion regarding this in **Supplementary Information: Degradation Time Length** (page 14).

We take a look at how long it takes to reach maximum PCE, and the result is shown in **Supplementary Fig. 10**. The majority of the dataset reaches maximum PCE before 150 hours (~96.8%, 2,245 ageing curves out of 2,320; the dataset in revised manuscript only includes solar cells degraded under N₂ atmosphere). We now take this into account and take out data points with maximum PCE reached beyond 150 hours. The analysis in the manuscript has been redone for **Fig. 1, 2, 4, and 5** and the number of data points in the manuscript have been updated. Despite the slightly smaller dataset, the conclusion has not been changed: (1) we still observe the decrease in ΔPCE_{rel} as the maximum PCE group mean increases (**Fig. 2**) and (2) the clustering of ageing curves generates the same groups and shapes (**Fig. 4**).

3. In Figure S3, what does the quantization error mean and how is it calculated? How is the optimum value $n=4$ determined?

A detailed explanation regarding how the optimum cluster value of $n = 4$ is determined is added in the **Supplementary Information: SOM Quantisation Error** (page 15).

4. In Figure 4, the degradation curve shapes are categorised into four different clusters containing one initial gain cluster and three exponential decay clusters. In some previous reports (10.1016/j.joule.2018.05.005 & 10.1038/s41560-019-0529-5 & 10.1038/s41560-017-0060-5), the burn-in period is independent of the linear degradation period and reversible, and the shape may be affected by device structures (initial decay for n-i-p and gain for p-i-n). The authors should check whether degradation curve shapes are related to device structures.

The Reviewer raises an important point. We have checked and run the algorithm for the two device structures, and the following results have been added in **Supplementary Information: Device Architecture Impact on Clustering** (page 18). Although the results for *p-i-n* and *n-i-p* architectures are statistically different, the general structure of the clusters appears to be similar. Both *p-i-n* and *n-i-p* devices can show initial decay and initial gain behaviours, but the percentages are slightly different.

5. The authors could also add a discussion on whether different degradation rates of the three exponential decay clusters are related to the testing conditions (temperature and atmosphere) since the device stability is sensitive to them.

The Reviewer raises an important point regarding the dependency of clusters with the testing conditions. A detailed explanation has been added to the **Supplementary Information: Correlation between Clusters and Testing Conditions** (page 19). This time, we exclude the cells aged under air atmosphere (48 data points), resulting in total 'cleaned' data points of 2,245, because the cells' number is too low.

Reviewer #2 (Remarks to the Author):

Machine learning, one of artificial intelligence methods is useful in many fields. In this manuscript, the relationship between efficiency and stability of perovskite solar cells (PSCs) is explained from a novel perspective, i.e. using the unsupervised machine learning method self-organising map. Unsupervised clustering is achieved according to data topology information. Moreover, the clustering results are good, which can support the conclusions. The novelty of this work is high compared with the traditional additive strategy in PSCs. Therefore, I suggest publication. However, this paper is preliminary and needs major revision. The suggestions are followed:

1. First of all, this paper is not well organized. Too many details should be put into supporting information such as Dataset description. The authors should give logical description in the Dataset description: how we get higher data quality? Tell the readers how high the data quality used in this paper. By the way, is artificial intelligence used for the paper writing, such as ChatGPT?

We thank the reviewer for the constructive criticism. Yet we consider a brief description of the dataset as useful to the reader, which is why we left a briefer version of it in the main manuscript.

We have added a brief discussion of the data quality of our dataset into **Supplementary Information: Data Quality** (page 20).

Regarding the paper-writing process, artificial intelligence, including ChatGPT, was not used.

2. This manuscript only compared the results of SOM and k-means, as well as analysis of different k values of k-means. However, it lacks direct analysis on the number of SOM clusters.

The selection of optimum number of clusters has been elaborated in the answer for Reviewer 1, Question 3 above and added to **Supplementary Information: SOM Quantisation Error** (page 15) and **K-means Clustering** (page 20).

3. The figures are not satisfactory since researches should consider the art of science. Figure 1(a) should be from the actual experimental data. What is the exact value in the longitudinal coordinate in Figure 1a?

We have updated **Fig. 1a** in the main text of the manuscript (page 5) as the following to give an example of the actual data.

Fig. 1: Statistical analysis of a large MPPT ageing dataset. **a** Schematic on how the relative change between maximum PCE and PCE at 150 hours ($\Delta PCE, rel$) is calculated. **b** The $\Delta PCE, rel$ for the five groups of maximum PCE. The solid line in the box represents the median, the dashed line in the box represents the mean, the error bar represents the corresponding interquartile range (between the 25th and 75th percent quartile), and the shaded area represents the distribution of the data points for the group.

4. The theory from Figure 3 is common sense. Authors should consider the detailed physical process in perovskite solar cells. How excess energy triggered the degradation in physics? This will help readers to connect machine learning results with the physical process in perovskite solar cells.

We pointed out that it is impossible to directly relate the results of the ML analysis to the physical causes of degradation, since many different layers and stacks are present in the dataset with likely a large variation on degradation patterns. This is why we wanted to provide a “high-level” explanation for the high-level analysis. We have refined the explanation of our model and hope that it is satisfactory now, and updated it in the main text (page 7, line 213-216).

Reviewer #3 (Remarks to the Author):

Synopsis

“Stability Follows Efficiency: Analysis of a Large Perovskite Solar Cells Ageing Dataset” by Hartono et al. performs interesting statistical analyses on degradation data collected from several thousand perovskite solar cells over a period of years. The works highlights several interesting correlations between the initial efficiency performance of solar cells, the rate at which the degrade,

and the “shape” with which their PCE vs. time curves change. Rather than relying on any specific information about the cells, like their architecture, transport layers, absorber, testing temperature, or even sub-degradation type (FF, Jsc, Voc) to understand how and why they degrade, the work focuses purely on high-level abstraction of trends across thousands of cells that vary in essentially every way that perovskite solar cells can vary. Intuitively, I suspect that by zooming in on the degradation difference between any pair of solar cells in the work, the reasons why they degrade differently would be apparent from things like material choice. However, I think there is value also in this high-level approach, since it may help the community identify larger trends. Thus, after the points below are addressed, I think the work is suitable for publication in Nature Communications.

Major Points

1. The definition of TS80m is not clear from the description provided from lines 53-55. The definition of Ts80 is clear, but in saying “...normalizes an extrapolated TS80 ... to a reference condition...”, it is unclear what normalization is being performed. Normalizing to efficiency at a reference condition? Using an acceleration-factor like analysis to predict lifetime at the reference condition? This should be clarified.

The paragraph explaining T_{S80m} has been edited in page 3, line 50-58 to increase clarity of the figure of merit, as following.

Zhang et al. recently performed a statistical analysis on the Perovskite Database Project¹ by introducing a new figure of merit for stability called T_{S80m} . They project the measured T_{S80} (the time taken to reach 80% of the stabilised efficiency at the end of the burn-in region)² to T_{S80m} , which is the predicted value under reference conditions (300K, 20% relative humidity, and 1 sun illumination). The authors re-calculate T_{S80} with the help of acceleration factors which are determined based on various assumptions and which consider the temperature, humidity, and illumination levels during the actual ageing test.²

2. On lines 88-89, the average number of batches contributed per researcher (or a distribution of batch numbers) is also of interest. Since the “minimum” number of solar cells per batch is 48, and the average number of solar cells tested per researcher is 72, this description could be interpreted as many (or most) researchers contributing just one large batch to test stability, or as many (or all) researchers contributing just a few cells per batch for lifetime testing in which case each researcher would have cells from perhaps 10+ batches on average being tested.

We thank the reviewer for this useful comment. The average number of solar cells per researcher can unfortunately not directly be calculated from the number of researchers and the number of data points in the analysis. The reason is that the data preparation procedure includes the removal of invalid (e.g. non-contacted) and noisy/faulty MPP-tracks. The full procedure is explained in the article on the High-Throughput Ageing System by Köbler et al.³ This is necessary due to the low reproducibility of PSCs in general⁴ and will lead to a reduced number of analysed cells compared to the number of cells produced per researcher. We therefore removed the “average number of cells tested per researcher” from the manuscript since it is misleading and irrelevant to the study.

3. Since care is given to define Maximum PCE (which may either arrive at $t = 0$, or slightly after in the case of cells with a burn-in), similar care should be given to Minimum PCE over the course of the test. Equation 1 is defined for a non-general case, where the Minimum PCE universally occurs at the end of the testing period explored (150 hr). However, if a cell increases in PCE and does not decay within the 150 hr window, then Minimum PCE would occur at $t = 0$ rather than $t = 150$ and thus the relative change in PCE could be written more generally as $(\text{Maximum PCE} - \text{Minimum PCE}) / \text{Maximum PCE}$. Perhaps all of the solar cells within the data set decay continuously and thus reach their lowest point at 150 hr, but if this is the case, it should be stated explicitly.

We thank the reviewer for this useful comment. We added a detailed discussion on the determination of the stability figure of merit ΔPCE_{rel} in **Supplementary Information: Degradation Time Length** (page 14). Additionally, we have removed MPP-tracks from the analysis for which the maximum PCE is reached beyond 150 hours, which mainly eliminates the concern of a “flipped” delta PCE where the minimum is at $t = 0$. In a special case of the initial gain type curve, it is still possible that the minimum is not at 150 hours but at $t = 0$ (see **Supplementary Fig. 3b**). Yet, we kept the previous definition of ΔPCE_{rel} rather than using the general case definition suggested by the reviewer. The reason is that we want to analyse the efficiency after a specific time of *operation* rather than evaluating the minimum PCE during the tracking time.

4. The concept described in lines 160 – 168 that more efficient cells dissipate energy in positive ways, and thus leave less excess loss energy to drive degradation is an interesting concept. However, the given dataset would need considerably more information to justify this speculation. In particular, it is more interesting to know what percentage of the absorbed energy is converted into electricity rather than the percentage of incident energy (which is PCE, by definition). Otherwise, differences in PCE could be a function solely of absorbing more energy, even if the conversion efficiency of absorbed power is the same across device architectures and efficiency groups. For example, if the internal conversion efficiency of all cells was 70%, and differences in PCE were solely from absorption, the higher PCE cells would actually have a larger amount of excess energy remaining in the cell than the lower efficiency cells. An estimation or measurement of the internal conversion of the tested cells would be an interesting addition to the work.

We recognize that the efficiency of a cell depends on its band gap, which corresponds to its open-circuit voltage limit and efficiency limit. We added a detailed discussion on the effect of perovskite absorbers’ band gaps on the maximum energy absorbed and the energy left for initiating degradation in dependency of the absolute efficiency in **Supplementary Information: Influence of band gaps** (page 8).

5. The statement beginning on line 175 that at JSC and VOC “no energy is extracted from the system” should be more nuanced, since by thermodynamic definition, ~100% of the energy is extracted from the system at JSC and VOC, but is done via re-radiation and thermal conduction rather than by providing an external circuit with electrical power. The discussion from lines 170 to 184 should also include the effect that voltage itself may have on perovskites.

We have considered the valuable input by the reviewer and refined the paragraph (page 7, line 194-200), providing an explanation regarding this process.

The reasoning is that at MPP, the maximum possible amount of electric energy is extracted from the device, leaving less energy or charges in the device potentially triggering degradation, which leads to the higher stability observed under MPP-tracking compared to J_{SC} (short-circuit current) and V_{OC} (open-circuit voltage) conditions. At J_{SC} and V_{OC} conditions, no electric energy is extracted from the system and all absorbed energy is potentially available to trigger degradation processes. Note that under V_{OC} condition additional effects might come to play⁶⁻⁷.

Minor format and style suggestions

1. The abstract at least in the version I saw is not a continuous paragraph. There are two newline commands. It should be a continuous single paragraph

We have removed the newlines.

2. On line 45, “only” and “less than” are redundant, use one or the other.

We removed “only” (line 43).

3. The use of et al is applied inconsistently—for example, Graniero et al uses only the author’s family name, while Zhuang Zhang et al. uses both given and family name.

We have changed it to use the author’s family name (line 44).

4. “High-throughput Ageing System” need not be capitalized on line 62, unless it is a specific product or system with a title.

Since it is a specific system³ we prefer to have it capitalised (line 62).

5. On line 97, the formula for the 3CAT perovskite should include the stoichiometry co-dependencies of x, y, z, m, and n (e.g. $CsxMAyFAzPbImBrn$, where $x + y + z = 1$ and $m + n = 3$)

We have updated the formula for 3CAT perovskite (line 95-96).

References

1. Jacobsson, T. J. *et al.* An open-access database and analysis tool for perovskite solar cells based on the FAIR data principles. *Nat. Energy* **7**, 107–115 (2022).
2. Zhang, Z., Wang, H., Jacobsson, T. J. & Luo, J. Big data driven perovskite solar cell stability analysis. *Nat. Commun.* **13**, 7639 (2022).

3. Köbler, H. *et al.* High-Throughput Aging System for Parallel Maximum Power Point Tracking of Perovskite Solar Cells. *Energy Technol.* **10**, 2200234 (2022).
4. Goetz, K. P. & Vaynzof, Y. The Challenge of Making the Same Device Twice in Perovskite Photovoltaics. *ACS Energy Lett.* **7**, 1750–1757 (2022).
5. Khenkin, M. V., M, A. K., Katz, E. A. & Visoly-Fisher, I. Bias-dependent degradation of various solar cells: lessons for stability of perovskite photovoltaics. *Energy Environ. Sci.* **12**, 550–558 (2019).
6. Prete, M. *et al.* Bias-Dependent Dynamics of Degradation and Recovery in Perovskite Solar Cells. *ACS Appl. Energy Mater.* **4**, 6562–6573 (2021).
7. Kim, D. *et al.* Light- and bias-induced structural variations in metal halide perovskites. *Nat. Commun.* **10**, 444 (2019).

REVIEWER COMMENTS

Reviewer #1 (Remarks to the Author):

The authors have addressed my comments, and the manuscript can be considered for publication now.

Reviewer #2 (Remarks to the Author):

Most of the comments from all three reviewers are well concerned. More works from the combination of perovskite and machine learning are expected to be published in the future. This work can attract interests from materials science and computer science. I think this work deserves publication now.

Reviewer #3 (Remarks to the Author):

Overall, the manuscript is improved, and I am left with just one final comment which should be addressed prior to publication:

In the new discussion of energy losses and degradation beginning on line 172,, the statement "according to the detailed balance limit of efficiency by Shockley-Queisser, this leaves a maximum of ~30% of the input energy to be absorbed by a solar cell under AM1.5G for a bandgap of 1.6-1.7 eV corresponding to the "triple cation" perovskite" is highly inaccurate. This statement conflates the theoretical maximum power conversion efficiency of a 1.6-1.7 eV bandgap cell (~30% according to the S-Q limit) with "maximum input energy". In contrast to the author's new claim, a cell with a 1.6-1.7 eV bandgap can absorb 50-60% of the total solar energy (since that is fraction of solar energy available at energies >1.6 eV), then a significant amount of this energy will be dissipated to produce a much lower power conversion efficiency (e.g. 30% at the S-Q limit, or 10-20% in the PSCs explored). This highlights the challenge in making the argument the authors are trying to make, since even in extremely good cells, ~50% of the absorbed energy is dissipated. This means that the contrast in energy dissipation between a 20% cell and a 10% cell (assuming they have identical absorption) is not nearly as large as the values claimed on lines 179 - 182. This entire discussion should be reworked including the relative size of the arrows coming out of Fig. 3.

REVIEWER COMMENTS

Reviewer #1 (Remarks to the Author):

The authors have addressed my comments, and the manuscript can be considered for publication now.

We thank the reviewer for the valuable inputs throughout the peer-review process.

Reviewer #2 (Remarks to the Author):

Most of the comments from all three reviewers are well concerned. More works from the combination of perovskite and machine learning are expected to be published in the future. This work can attract interests from materials science and computer science. I think this work deserves publication now.

We thank the reviewer for the valuable inputs throughout the peer-review process.

Reviewer #3 (Remarks to the Author):

Overall, the manuscript is improved, and I am left with just one final comment which should be addressed prior to publication:

In the new discussion of energy losses and degradation beginning on line 172,, the statement “according to the detailed balance limit of efficiency by Shockley-Queisser, this leaves a maximum of ~30% of the input energy to be absorbed by a solar cell under AM1.5G for a bandgap of 1.6-1.7 eV corresponding to the “triple cation” perovskite” is highly inaccurate. This statement conflates the theoretical maximum power conversion efficiency of a 1.6-1.7 eV bandgap cell (~30% according to the S-Q limit) with “maximum input energy”. In contrast to the author’s new claim, a cell with a 1.6-1.7 eV bandgap can absorb 50-60% of the total solar energy (since that is fraction of solar energy available at energies >1.6 eV), then a significant amount of this energy will be dissipated to produce a much lower power conversion efficiency (e.g. 30% at the S-Q limit, or 10-20% in the PSCs explored). This highlights the challenge in making the argument the authors are trying to make, since even in extremely good cells, ~50% of the absorbed energy is

dissipated. This means that the contrast in energy dissipation between a 20% cell and a 10% cell (assuming they have identical absorption) is not nearly as large as the values claimed on lines 179 – 182. This entire discussion should be reworked including the relative size of the arrows coming out of Fig. 3.

The Reviewer raises an important point, and we agree that we should have used the absorbed energy instead of the theoretical Shockley-Queisser limit. We have rewritten and clarified it as following:

*According to the detailed balance limit of efficiency by Shockley-Queisser, this leaves ~43-48 % of the input energy to be absorbed, and a maximum ~30% of the input energy to be converted into electric energy by a solar cell under AM1.5G for a band gap of 1.6-1.7 eV corresponding to the “triple cation” perovskite which provides the majority of data points of the present data. Assuming an equal or similar band gap and similar optical properties, the differences between a low and a high efficient solar cell originate in recombination¹² or transport losses¹³. The fraction of absorbed energy, which is not extracted again from the device electrically, remains in the solar cell and gets dissipated in the device where it can potentially trigger degradation. For example, in a 10% efficient “triple cation” (energy absorbed ~466 W/m²) device, ~366 W/m² of the input energy would be potentially available to cause damage, under 1 sun, while for a 20% device it would be ~266 W/m². For an estimation of the influence of the band gap in this dataset^{9,14-21} on this theory, see **Supplementary Information: Influence of band gaps, Supplementary Table 3, Supplementary Figs. 4 and 5.***

The discussion in the **Supplementary Information: Influence of band gaps** has also been updated.

Fig. 3 is a schematic figure, which has been revised to the following, to ‘separate’ the energy losses (e.g. transmission), and the ‘excess energy’ potentially triggering degradation.

Fig. 3: The energy conservation schematic for the perovskite solar cell system. The incident solar power coming into the system equals the power coming out. In the case of high-efficiency cells, the actual power extracted and converted into electricity is relatively large, leaving a smaller amount of energy to potentially trigger degradation.